# High glucose levels increase influenza-associated damage to the pulmonary epithelial-endothelial barrier

Katina D Hulme[1], Limin Yan[1], Rebecca J Marshall[1], Conor J Bloxham[2], Kyle R Upton[1], Sumaira Z Hasnain[3], Helle Bielefeldt-Ohmann[1,4], Zhixuan Loh[5], Katharina Ronacher[3,4], Keng Yih Chew[1], Linda A Gallo[2,3], Kirsty R Short[1,4]*

[1]School of Chemistry and Molecular Biosciences, The University of Queensland, St Lucia, Australia; [2]School of Biomedical Sciences, The University of Queensland, Woolloongabba, Australia; [3]Mater Research Institute, The University of Queensland, Translational Research Institute, Woolloongabba, Australia; [4]Australian Infectious Diseases Research Centre, The University of Queensland, St Lucia, Australia; [5]Institute for Molecular Bioscience, The University of Queensland, St. Lucia, Australia

**Abstract** Diabetes mellitus is a known susceptibility factor for severe influenza virus infections. However, the mechanisms that underlie this susceptibility remain incompletely understood. Here, the effects of high glucose levels on influenza severity were investigated using an *in vitro* model of the pulmonary epithelial-endothelial barrier as well as an *in vivo* murine model of type II diabetes. *In vitro* we show that high glucose conditions prior to IAV infection increased virus-induced barrier damage. This was associated with an increased pro-inflammatory response in endothelial cells and the subsequent damage of the epithelial junctional complex. These results were subsequently validated *in vivo*. This study provides the first evidence that hyperglycaemia may increase influenza severity by damaging the pulmonary epithelial-endothelial barrier and increasing pulmonary oedema. These data suggest that maintaining long-term glucose control in individuals with diabetes is paramount in reducing the morbidity and mortality associated with influenza virus infections.

*For correspondence:
k.short@uq.edu.au

## Introduction

Every year influenza A virus (IAV) infects 5–15% of the world's population (*Goeijenbier et al., 2017*; *Stöhr, 2002*). Typically, influenza virus causes an acute and self-limiting disease. However, influenza virus can cause severe disease in individuals with certain chronic medical conditions, including diabetes mellitus (*Short et al., 2014*). Upon infection with influenza virus, patients with diabetes have triple the risk of hospitalisation, quadruple the risk of admission to the intensive care unit and double the risk of a fatal outcome compared to individuals with no underlying illness (*Allard et al., 2010*; *Wilking et al., 2010*). However, the mechanisms that drive this susceptibility remain incompletely understood.

Diabetes is characterised by chronically high blood glucose levels (hyperglycaemia) and is separated into two primary categories: type I (T1D) and type II (T2D). T1D accounts for ~10% of cases (*Chatterjee et al., 2017*) and is generally regarded as an autoimmune disorder in which the immune system attacks the insulin-producing pancreatic β-cells (*Brody, 2012*; *Federation ID, 2015*). The remaining ~90% of cases belong to the T2D category which is characterised by relative insulin deficiency as a result of pancreatic β-cell dysfunction and insulin resistance (*Chatterjee et al., 2017*; *Defronzo, 2009*). Diabetes is associated with complications that affect multiple organ systems

(*Abdel-Rahman et al., 2012*). Notably, when compared to healthy individuals, people with diabetes have a two- to fourfold increased risk of cardiovascular disease (*Rao Kondapally Seshasai et al., 2011*). Similarly, elevated blood glucose levels in patients with diabetes are associated with kidney damage and the development of diabetic kidney disease (*Carpentier et al., 2018*).

In addition to being associated with the micro- and macrovascular complications of diabetes, hyperglycaemia is also associated with the increased susceptibility to infectious diseases (*Critchley et al., 2018*). For example, in a Danish population, each 1 mmol/L increase in plasma glucose level was associated with a 6% increased risk of hospitalisation for pneumonia (*Benfield et al., 2007*). In the specific case of IAV, hyperglycaemia may increase disease severity via either; immunosuppression (*Reading et al., 1998*), increased epithelial infection (*Kohio and Adamson, 2013*) and/or, altered activity of transporters responsible for keeping the lung free of oedematous fluid (*Yamashita et al., 1992*; *Rivelli et al., 2012*).

Thaiss and colleagues recently showed that hyperglycaemia contributes to the elevated severity of enteric infections in patients with diabetes by increasing the permeability of the gut epithelium (*Thaiss et al., 2018*). Specifically, GLUT2, a bidirectional glucose transporter, was identified as a key driver of dysfunction of the epithelial junctional complex (*Thaiss et al., 2018*). In addition to their role in the gut, the epithelial junctional complex plays an important role in the lung by limiting the amount of fluid present in the alveolar airspace. Junctional proteins are formed between both pulmonary epithelial cells and pulmonary endothelial cells, although the epithelial cell component plays a more significant role keeping the lung free of fluid (*Short et al., 2014*). We have previously demonstrated that IAV can damage junctional proteins between adjacent human alveolar epithelial cells and therefore impair efficient gas-exchange (*Short et al., 2016*). The findings of Thaiss and colleagues (*Thaiss et al., 2018*) raise the intriguing possibility that IAV-induced damage to epithelial tight junctions is further exacerbated in the context of hyperglycaemia and that this may account for the increased severity of influenza seen in patients with diabetes.

Here, we use an *in vitro* co-culture model of the pulmonary epithelial-endothelial barrier and a murine model of T2D to provide new insights into the role of hyperglycaemia and IAV-induced barrier damage in the lung. We show that compared to normal glucose concentrations, elevated glucose levels prior to IAV infection increase virus-induced barrier damage, both *in vitro* and *in vivo*, and that this damage is associated with endothelial-driven inflammation.

## Results

### A history of high glucose increases influenza-induced barrier damage in an *in vitro* epithelial-endothelial co-culture

To determine the role of high glucose levels (representative of hyperglycaemia) in IAV-induced barrier damage, we adapted our previously described model of the human alveolar epithelial-endothelial barrier (*Short et al., 2016*; *Figure 1A*). This is a model of the distal lung region where, *in vivo*, epithelial cells are entirely covered by a fluid layer, known as the alveolar lining fluid, to facilitate efficient gas exchange (*Fronius et al., 2012*). As such, in the *in vitro* model of this region the cells are cultured in medium rather than at an air-liquid interface, which is more appropriate for upper airway models. To mimic normal or hyperglycaemic conditions, respectively, the medium of the lower co-culture compartment contained either 7 mM glucose or 12 mM glucose (*Figure 1A*). These concentrations were selected as these reflect clinically relevant glucose levels (i.e. >7 mM glucose is considered diabetic) (*Danaei et al., 2011*; *Expert Committee on the Diagnosis and Classification of Diabetes Mellitus, 2003*). Cells differentiated within 3–5 days of culture as previously characterised within this co-culture model (*Hermanns et al., 2004*). The development of an elevated TEER was not affected by the differing glucose concentrations (*Figure 1—figure supplement 1*), and culture of primary endothelial cells in RPMI media did not affect their endothelial phenotype (*Figure 1—figure supplement 2*) as has been previously described (*Short et al., 2016*). IAV was then added to the upper compartment to mimic IAV infection via the respiratory route. Importantly, infections of all treatment groups were performed in the presence of 12 mM glucose (in both the upper and lower compartment) to mimic the fact that upon physiological stressors, such as IAV infection, individuals typically experience elevated blood glucose levels (*Marik and Bellomo, 2013*; *Karlsson et al., 2019*). Barrier integrity was then compared between co-cultures with a history of elevated (12 mM)

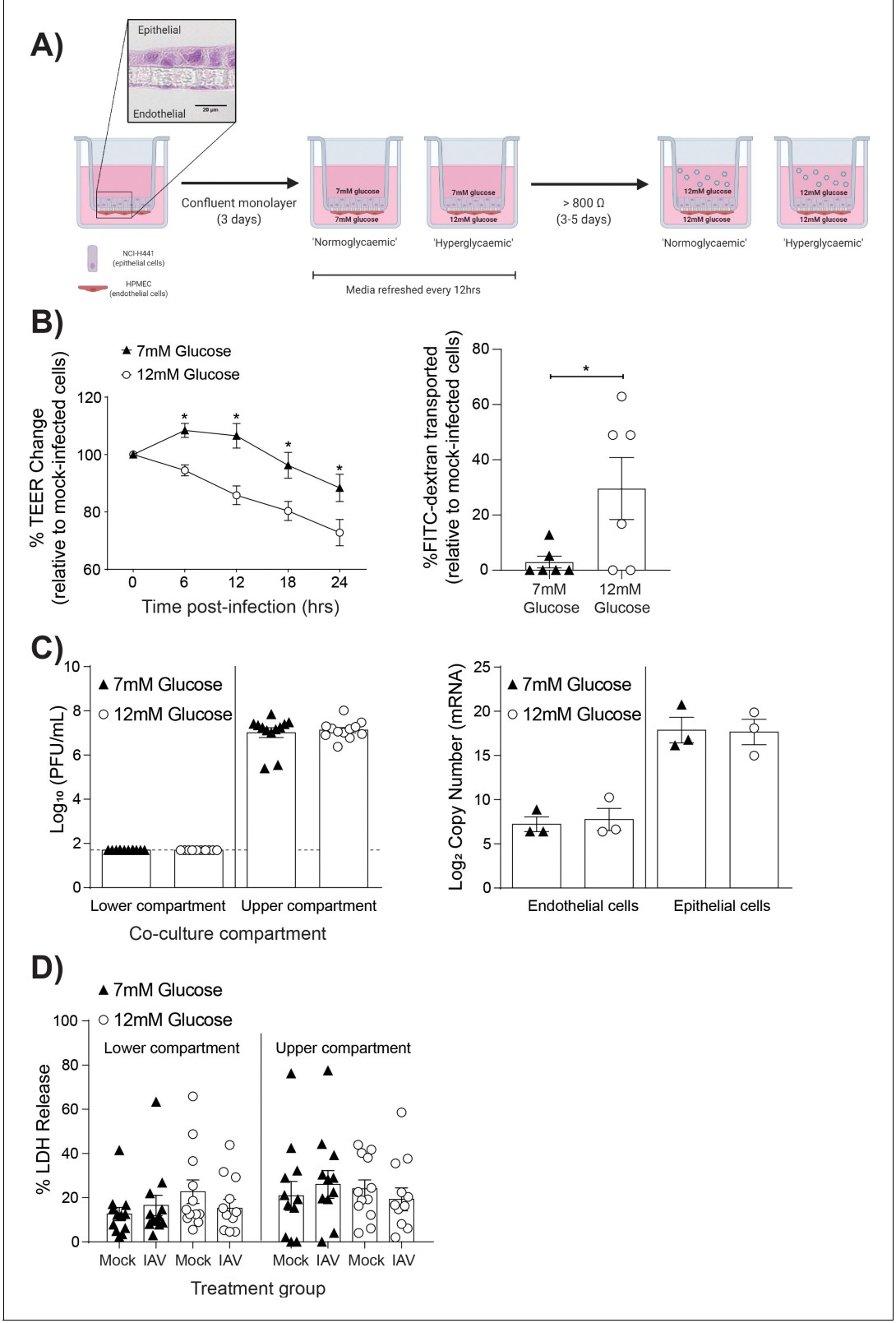

**Figure 1.** High glucose conditions increase IAV-induced barrier independent of cell death. (**A**) Schematic representation of the *in vitro* co-culture model of the alveolar epithelial-endothelial barrier. Image created with Biorender. Micrograph (40x magnification) of transwell membrane. (**B**) Left: Measurement of co-culture barrier integrity using TEER ($\Omega$) readings following infection with medium ('mock') or IAV. Data are expressed relative to the baseline TEER and the TEER of mock-infected cells at each time point. Statistical significance was determined using a two-way ANOVA with a

*Figure 1 continued on next page*

*Figure 1 continued*

Bonferroni post-test. Right: Permeability of co-cultures to FITC-dextran 24 hr post-infection. Data shows the percentage of FITC detected in the lower compartment relative to mock-infected wells (defined as 0). Statistical significance was determined using a Student's t-test. (C) Left: PFU/mL of IAV detected in the lower and upper compartment of the co-culture 24 hr post-IAV infection. A dashed line indicates the detection limit of the assay. Right: mRNA detected by qPCR in epithelial and endothelial cells 24 hr post-IAV infection. Viral replication represented as viral copy number. Statistical significance was determined using a Student's t-test. (D) Percentage release of LDH from co-cultured epithelial (upper compartment) and endothelial (lower compartment) cells at 24 hr post-infection. Statistical significance was determined using a Student's t-test. All data are pooled from a minimum of three independent experiments (with six biological replicates per group) and are shown as mean ± SEM. *: p<0.05.

The online version of this article includes the following figure supplement(s) for figure 1:

**Figure supplement 1.** TEER (Ω) of epithelial-endothelial co-cultures following 3–5 days of different glucose concentrations in the lower compartment of the transwell.
**Figure supplement 2.** Tube formation of HPMECs under different media conditions.

glucose levels (henceforth referred to as 'high glucose co-cultures') and a history of healthy (7 mM) glucose (levels henceforth referred to as 'low glucose co-cultures') Strikingly, following IAV infection, high glucose co-cultures had a significantly greater decrease in TEER compared to low glucose co-cultures (*Figure 1B*). These data were consistent with the increased permeability to FITC-labeled dextran observed in infected high glucose co-cultures relative to infected low glucose co-cultures (*Figure 1B*).

Previously, it has been suggested that high glucose conditions increase IAV replication *in vitro* (*Kohio and Adamson, 2013*). Therefore, the possibility that the observed barrier damage in the high glucose co-cultures was the result of increased viral replication was explored. However, no statistically significant difference in viral messenger RNA (mRNA) or infectious virus in supernatant was observed between the different glucose conditions (*Figure 1C*).

Human umbilical vein endothelial cells exposed to high glucose concentrations have an increased level of apoptosis compared to cells in lower glucose concentrations (*Risso et al., 2001*). Therefore, we next investigated whether the observed barrier damage in the infected high glucose co-cultures was the result of increased cell death via the crude measure of lactate dehydrogenase (LDH) release. While LDH levels cannot definitively rule out a role of cell death, no significant difference was observed in cells from the high and low glucose co-cultures (*Figure 1D*).

Together, these data suggest that a history of high glucose increases the IAV-induced damage to the epithelial-endothelial barrier *in vitro* in the absence of wide-spread cell death or increased viral replication.

## High glucose barrier damage was dependent upon the presence of endothelial cells

To assess if endothelial cells contributed to the observed barrier damage, mono-cultures of epithelial cells were established and subjected to the same experimental protocols as the co-cultures (*Figure 2A*). Importantly, as epithelial cells constitute the main component of the pulmonary epithelial-endothelial barrier, mono-cultures still displayed an elevated TEER after 4–5 days in culture (data not shown). However, endothelial cells provided a stabilising effect on the integrity of the epithelial barrier (*Figure 2—figure supplement 1*), with co-cultures having a significantly higher pre-infection TEER when compared to epithelial mono-cultures. Accordingly, mono and co-cultures were not directly compared; rather mono-cultures with 7 mM glucose were compared to mono-cultures with 12 mM glucose. Following IAV infection, infected high glucose mono-cultures showed no statistically significant decrease in TEER overtime compared to infected low glucose mono-cultures (*Figure 2B*). Similarly, no difference in epithelial permeability to FITC-labelled dextran was observed (*Figure 2B*). Consistent with these data, no significant difference in viral replication (*Figure 2C*) nor LDH release was observed (*Figure 2D*). Thus, given that high glucose IAV-induced barrier damage was not observed in the absence of endothelial cells, these data suggest that endothelial cells are the main drivers of barrier damage observed in this *in vitro* model.

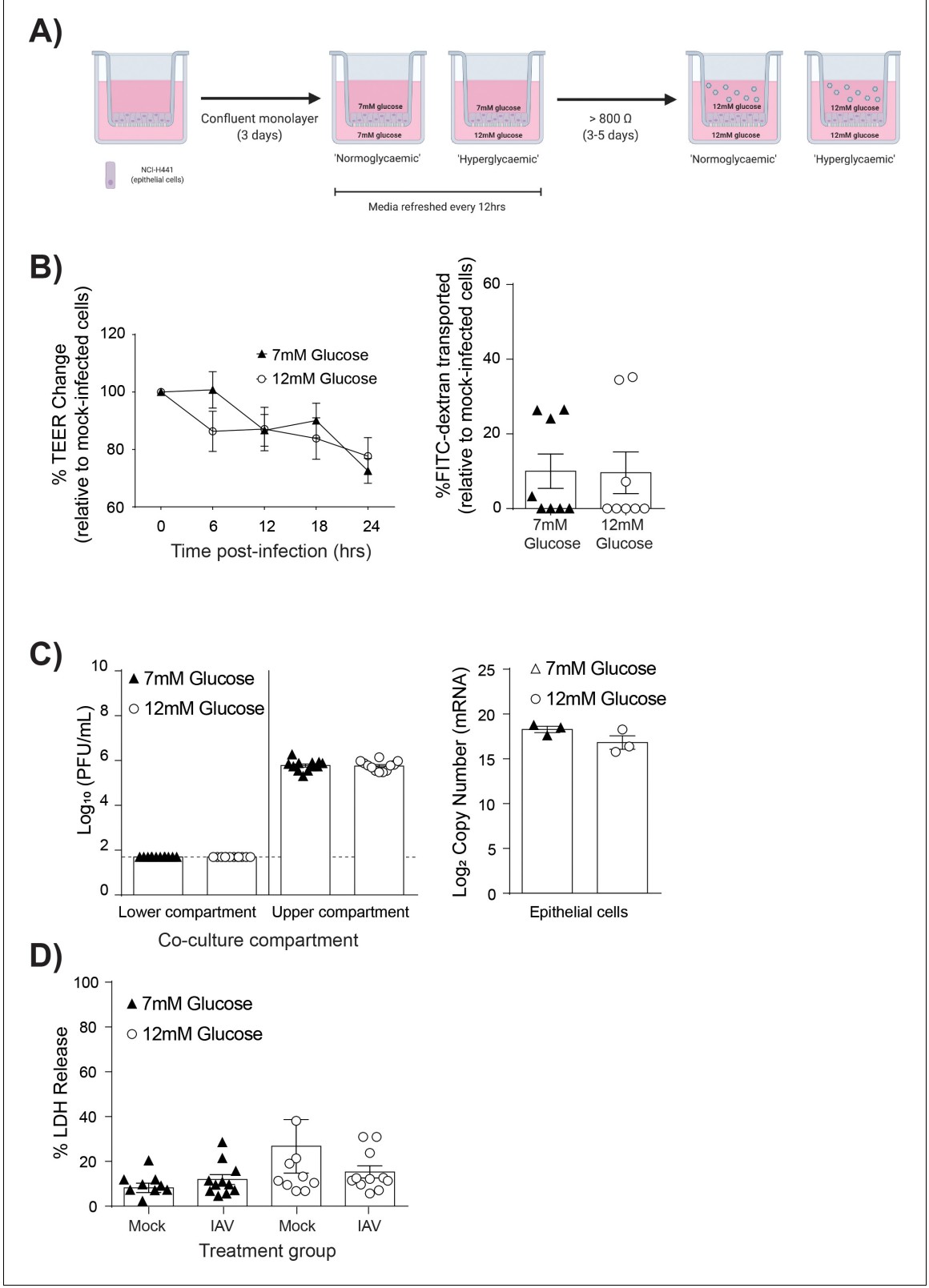

**Figure 2.** IAV-induced barrier damage in high glucose condition is dependent upon the presence of endothelial cells. (**A**) Schematic representation of the *in vitro* mono-culture model of the alveolar epithelial-endothelial barrier. Image created with Biorender. (**B**) Left: Measurement of epithelial mono-culture barrier integrity using TEER ($\Omega$) readings following infection with medium ('mock') or IAV. Data are expressed relative to the baseline TEER and the TEER of mock-infected cells at each time point. Statistical significance was determined using a two-way ANOVA with a Bonferroni post-test. Right:
*Figure 2 continued on next page*

*Figure 2 continued*

Permeability of epithelial mono-culture to FITC-dextran 24 hr post-infection. Data show the percentage of FITC detected in the lower compartment relative to mock-infected wells (defined as 0). Infected wells that had less detected FITC than mock-infected were set to 0. Statistical significance was determined using a Student's unpaired t-test. (C) Left: PFU/mL of IAV detected in the lower compartment and upper compartment 24 hr post-IAV infection. A dashed line indicates the detection limit of the assay. Right: mRNA detected by qPCR in epithelial cells 24 hr post-IAV infection. Viral replication represented as viral copy. Statistical significance was determined using a Student's unpaired t-test. (D) Percentage release of LDH from mono-culture epithelial (upper compartment) cells at 24 hr post-infection. Statistical significance was determined using a Student's t-test. All data are pooled from a minimum of three independent experiments (with six biological replicates per group) and are shown as mean ± SEM. *: p<0.05.
The online version of this article includes the following figure supplement(s) for figure 2:

**Figure supplement 1.** Endothelial cells provide a stabilising effect on the integrity of the epithelial barrier.

## IAV-induced barrier damage in infected high glucose co-cultures is associated with a pro-inflammatory response in endothelial cells

We next sought to determine how endothelial cells contributed to the barrier damage observed in IAV-infected high glucose co-cultures. To this end, the transcriptome of endothelial cells from high and low glucose co-cultures was assessed using RNASeq. At 24 hr post-IAV infection, numerous differentially expressed genes were identified in endothelial cells derived from infected co-cultures relative to endothelial cells derived from uninfected co-cultures (*Figure 3A*). Many of these genes were differentially expressed after infection under both low and high glucose conditions (*Figure 3B*). Strikingly, certain pathways that were enriched in endothelial cells derived from infected, low glucose co-cultures were not enriched in endothelial cells derived from infected, high glucose co-cultures (*Figure 3C*). Namely, while both low and high glucose endothelial cells expressed pathways associated with 'defence response to virus' and 'immune system processes', high glucose endothelial cells lacked a concomitant increase in regulatory pathways otherwise seen in low glucose endothelial cells (*Figure 3C*). These data suggest that endothelial cells derived from infected high glucose co-cultures may be in a more pro-inflammatory state than their counterparts in low glucose conditions. Consistent with this hypothesis, endothelial cells from IAV-infected high glucose co-cultures expressed significantly higher levels of pro-inflammatory chemokine *CXCL8* compared to endothelial cells from IAV-infected low glucose co-cultures (*Figure 3D*). Similarly, we observed an increase in IL-6 and IL-8 protein levels in endothelial supernatant 24 hr post-infection (*Figure 3E*). To determine whether high glucose endothelial cells, in the presence of IAV, produced a soluble pro-inflammatory factor that damaged the epithelial-endothelial barrier, the basolateral supernatant from high and low glucose co-cultures was harvested. This supernatant was then transferred to uninfected high and low glucose co-cultures. The transfer of the cell culture supernatant from high-glucose, infected co-cultures was sufficient to induce barrier damage in uninfected cells when compared to normal glucose conditions (*Figure 3F*). Consistent with a heat-labile soluble protein (such as a cytokine) driving the observed phenotype, the transfer of heat-treated supernatant from high glucose conditions did not damage the epithelial-endothelial barrier (*Figure 3F*). Furthermore, infection using ultraviolet inactivated virus did not induce barrier damage, even under hyperglycaemic conditions, indicating that initially, active virus is required to induce a reduction in TEER (*Figure 3—figure supplement 1*). These data suggest that the observed TEER decrease is not due to the presence of viral PAMPS/antigen in the media as it has been previously demonstrated that vRNA and mRNA remains present up to 70°C (*Jonges et al., 2010*). Interestingly, the somewhat slower kinetics of the TEER drop after supernatant transfer indicates a possibility of indirect action of the cytokine(s); that is, cytokines could trigger epithelial cells in the co-culture to produce another cytokine that then in turn damages the barrier. IL-1β is known to increase interstitial epithelial barrier permeability (*Al-Sadi and Ma, 2007*). We therefore assessed the effect of introducing an IL-1 antagonist to the system. No difference in barrier integrity with and without Anakinra was observed (*Figure 3—figure supplement 2*), indicating that while the phenotype is likely due to a soluble factor, it is not IL-1 dependent. These results are to be expected due to the redundancy in innate host defence pathways for pathogen recognition (*Nish and Medzhitov, 2011*).

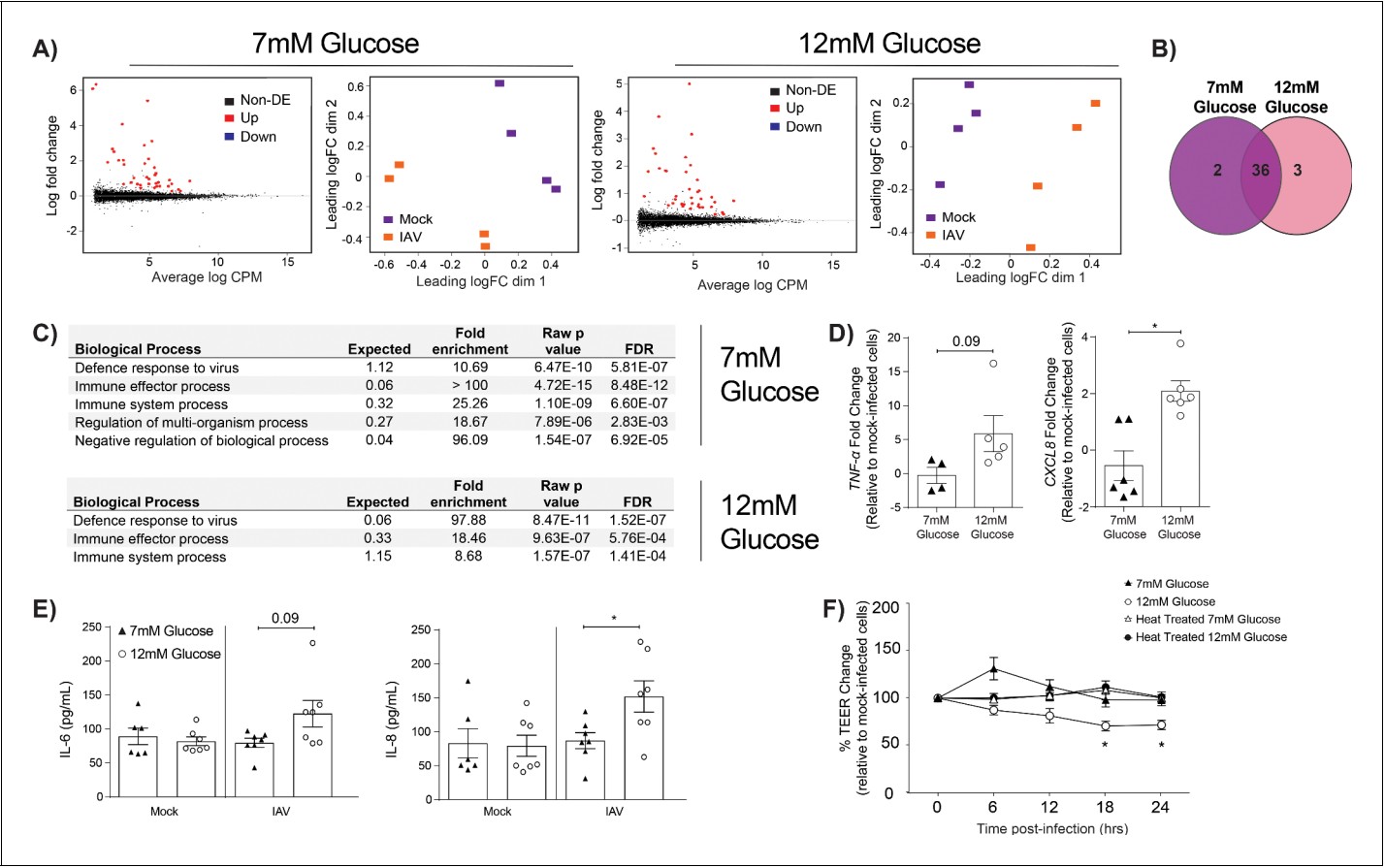

**Figure 3.** IAV-induced barrier damage in infected high glucose co-cultures is associated with a pro-inflammatory response in endothelial cells. (**A**) Left: Mean difference (MD) plot depicts the relationship between gene-wise average log expression and the log-fold change comparison between IAV and mock infected cells. DE = differentially expressed. Right: Multi-dimensional scaling (MDS) plot of the endothelial cell sample used for RNASeq shows distinct clustering of samples by treatment group. (**B**) The number of DE expressed genes that are detected in endothelial cells derived from IAV-infected co-cultures (data collected at 24 hr post-infection). (**C**) Biological processes that are enriched in the endothelial cells derived from IAV-infected co-cultures at 24 hr post-infection. (**D**) Pro-inflammatory gene expression in co-culture endothelial cells. Data are normalised to GAPDH expression and fold change was calculated using the ΔΔCt method, expressed relative to mock infected cells. Data are mean ± SEM of an average of two technical replicates per group from a minimum of three independent experiments. (**E**) Levels of cytokines in the lower compartment (endothelial) cell culture supernatant 24 hr post IAV infection. Statistical significance was determined using a Mann-Whitney test. *: p<0.05. (**F**) Measurement of co-culture barrier integrity using TEER (Ω) readings after the addition of supernatant to the lower compartment. Supernatant was derived from the lower compartment of mock-infected or IAV-infected co-cultures with a history of 7 mM or 12 mM glucose. Harvested media was either heat treated or transferred without heat treatment. Data are shown relative to the TEER measured before the addition of the cell culture supernatant and to the TEER of wells subject to mock-infected supernatant transfer at each time point, for each glucose condition. Data are shown as mean ± SEM of three independent experiments (with six biological replicates per group). Statistical significance was determined using a two-way ANOVA with a Tukey post hoc test. *: p<0.05.

The online version of this article includes the following figure supplement(s) for figure 3:

**Figure supplement 1.** Barrier damage is dependent on active virus.

**Figure supplement 2.** Barrier damage is and is not reversable via the addition of an IL-1 receptor antagonist.

## IAV-induced barrier damage in infected high glucose co-cultures is the result of an impaired epithelial junctional complex

A pro-inflammatory environment is known to facilitate the damage of the junctional complex between adjacent epithelial cells and increase barrier permeability (*Short et al., 2014*). Similarly, hyperglycaemia can damage the integrity of junctional complex between epithelial cells in the gut (*Thaiss et al., 2018*). We therefore assessed the integrity of the epithelial junctional complex in infected high and low glucose co-cultures. We elected to focus on the junctional proteins between adjacent epithelial cells, rather than endothelial cells, as the epithelium constitutes more than 90% of

resistance to protein transport across the epithelial-endothelial barrier and thus contributes the largest amount to the TEER in the co-culture system (*Short et al., 2014*). No difference in epithelial E-cadherin staining was observed between low and high glucose co-cultures (*Figure 4A*). However, infected high glucose co-cultures had significantly lower levels of Claudin-4 and JAM-1 between adjacent epithelial cells compared to infected low glucose co-cultures (*Figure 4B & C*). Consistent with our hypothesis that a soluble factor is driving the loss of barrier integrity, co-staining of Claudin-4 and the influenza A nucleoprotein demonstrate that barrier damage occurred in both infected and bystander cells (*Figure 4D*). To confirm that this damage of the junctional complex was dependent on endothelial cells, the same semi-qualitative analysis was performed on epithelial mono-cultures. In contrast to the co-cultures, there was no difference in junctional integrity of IAV-infected epithelial cells in low or high glucose mono-cultures (*Figure 5*). Taken together, these data suggest that under conditions of high glucose *in vitro*, active IAV triggers a pro-inflammatory response in endothelial cells which in turn damages the epithelial junctional complex.

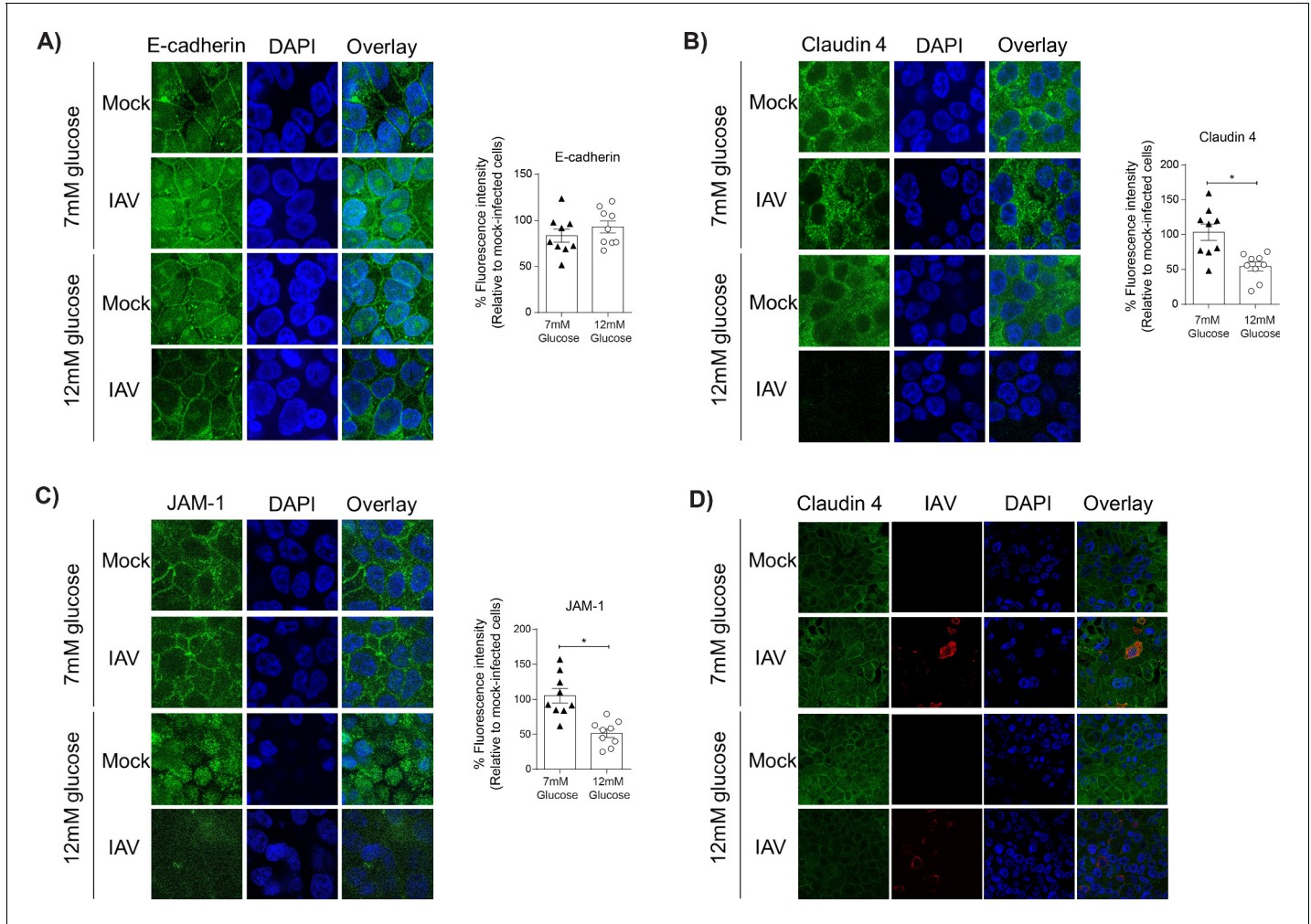

**Figure 4.** Barrier damage under high glucose conditions was associated with destruction of the apical junctional complex. (A–C) Left: Representative immunofluorescence images (63x magnification) of apical junction complex proteins of epithelial cells. Epithelial cells were grown on transwell membrane in co-culture with endothelial cells and infected with either medium ('mock') or IAV. At 24 hr post-infection, cells were fixed and the nucleus and the relevant tight junction proteins were stained (blue and green, respectively). Right: The percentage of fluorescence intensity in IAV-infected epithelial cells in co-culture relative to mock-infected cells (defined as 100%) at 24 hr post-infection. Statistical comparisons were made using a Student's unpaired t-test *: p<0.05. Data are pooled from three independent experiments (with three biological replicates per group) and shown as mean ± SEM. JAM-1: junctional adhesion molecule-1. (D) Representative immunofluorescence images (x63 magnification) of epithelial cells infected with either medium ('mock') or IAV. At 24 hr post-infection, cells were fixed and the nucleus, claudin 4 and influenza A nucleoprotein were stained (blue, green and red, respectively).

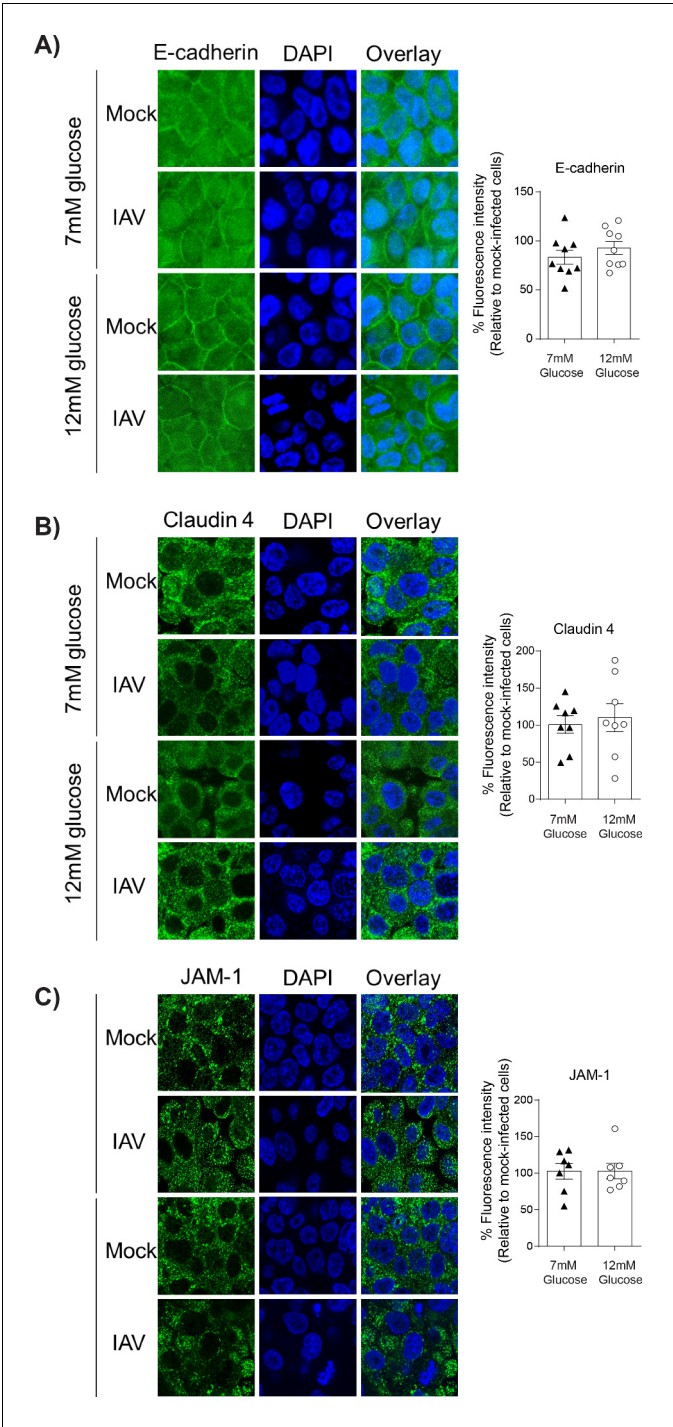

**Figure 5.** IAV-induced damage to the apical junctional complex is not observed in the absence of endothelial cells. **A–C)** Left: Representative immunofluorescence images (63x magnification) of apical junction complex proteins of epithelial cells. Epithelial cells were grown on transwell membrane in mono-culture and infected with either medium ('mock') or IAV. At 24 hr post-infection, cells were fixed and the nucleus and the relevant tight junction proteins were stained (blue and green, respectively). Right: The percentage of fluorescence intensity in IAV-infected epithelial cells in mono-culture relative to mock-infected cells (defined as 100%) at 24 hr post-infection. Statistical comparisons were made using a Student's unpaired t-test *: p<0.05. Data are pooled from three independent experiments (with three biological replicates per group) and shown as mean ± SEM. JAM-1: junctional adhesion molecule-1.

## High glucose levels *in vivo* are associated with more severe influenza and loss of tight junction integrity between pulmonary epithelial cells

Our *in vitro* findings are limited by constraints of the *in vitro* model, including the fact that only primary endothelial cells (and not primary epithelial cells) were employed. Thus, we next sought to validate our *in vitro* findings in a more complex *in vivo* model. To this end, we employed a well-established murine model of T2D where homozygote mice (*lepr*$^{db/db}$) are obese and hyperglycaemic whilst their heterozygote (*lepr*$^{db/+}$) counterparts are not (*Kodama et al., 1994*). Hyperglycaemia in *lepr*$^{db/db}$ mice was confirmed with a blood glucose test prior to infection (*Figure 6A*). Heterozygote and homozygote mice were subsequently infected with IAV and disease severity was monitored overtime. As demonstrated previously, weight-loss is not a clear marker for IAV disease severity of leptin receptor-deficient mice (*Radigan et al., 2014*). Consistent with these results, *lepr*$^{db/db}$ mice showed less weight loss than their *lepr*$^{db/+}$ counterparts (data not shown). It is well recognised that pulse-oximetry is a non-invasive measure of lung injury (*Lax et al., 2014*) that more accurately predicts disease severity than systemic weight loss in the case of murine influenza studies (*Verhoeven et al., 2009*). Moreover, blood oxygen saturation is a non-invasive measure of barrier damage – as when the epithelial-endothelial barrier is damaged gas exchange will occur less efficiently resulting in lower blood oxygen saturation. IAV-infected *lepr*$^{db/db}$ mice had significantly lower blood oxygen saturation at three days post-infection compared to *lepr*$^{db/+}$ mice (*Figure 6B*). Consistent with our *in vitro* data this was not associated with an increase in infectious virus present (*Figure 6C*); however, there was difference in viral messenger RNA (mRNA) between the homozygote and heterozygote mice (*Figure 6D*). Strikingly, and also consistent with our *in vitro* data, infected hyperglycaemic *lepr*$^{db/db}$ mice had a significant reduction in both JAM-1 and Claudin-4 levels between adjacent pulmonary epithelial cells compared to normoglycaemic *lepr*$^{db/+}$ mice (*Figure 6E–G*). Next, we measured pro-inflammatory cytokines in the lungs of infected normo- and hyperglycaemic mice after with IAV infection (*Figure 6H*). Consistent with our *in vitro* findings, IAV-infected *lepr*$^{db/db}$ mice had higher levels of pulmonary pro-inflammatory cytokines than infected *lepr*$^{db/+}$ mice, although this often just fell short of statistical significance (e.g. IP-10, p=0.06). Whilst we cannot offer definitive evidence that these cytokines were endothelial cell derived (as was the case with the *in vitro* data), these data are consistent with our *in vitro* data that loss of intact tight junctions in the context of IAV and hyperglycaemia is associated with a pro-inflammatory response.

## Discussion

It is well established that diabetes increases the risk of developing severe influenza (*Hulme et al., 2017*). However, the underlying mechanism for this observation remains to be fully elucidated. Here, we show that elevated glucose levels are associated with endothelial cell inflammation, destruction of the epithelial junctional complex and increased IAV severity. We therefore propose that in patients with hyperglycaemia, IAV triggers a pronounced inflammatory response in pulmonary endothelial cells. Secreted protein components of this inflammatory response (most likely cytokines) then damage the epithelial junctional complex and worsen IAV-induced barrier damage.

Hyperglycaemia has long been implicated in development of the various macro- and micro-vascular complications of diabetes. Similarly, hyperglycaemia has been implicated in the severity of both bacterial and viral infections in the respiratory tract (*Hulme et al., 2017*). Elevated blood glucose levels can directly increase glucose concentrations in airway secretions (*Philips et al., 2003*). Exposure of epithelial cells to elevated glucose concentrations then facilitates increased IAV replication by augmenting the assembly and activity of the cellular vacuolar-type H$^+$ ATPase (V-ATPase) (*Kohio and Adamson, 2013*). Interestingly, in the present study, we observed that hyperglycaemia increased IAV-induced barrier damage *in vitro* and increased influenza severity *in vivo* in the absence of increased viral replication (although initial viral replication was required to trigger epithelial barrier damage). This discrepancy may be the result of different disease models, IAV strains and/or time points used between studies. Similarly, IAV strain-dependent differences in pathogenesis, difference in cell culture glucose concentrations and the type of endothelial cell used may account for the inconsistencies between the results presented herein and those of our previous studies (*Short et al., 2016*).

Interestingly, previous studies have described elevated glucose levels can directly affect the activity of ATPases in epithelial cells (*Kohio and Adamson, 2013*; *Yamashita et al., 1992*; *Rivelli et al.,*

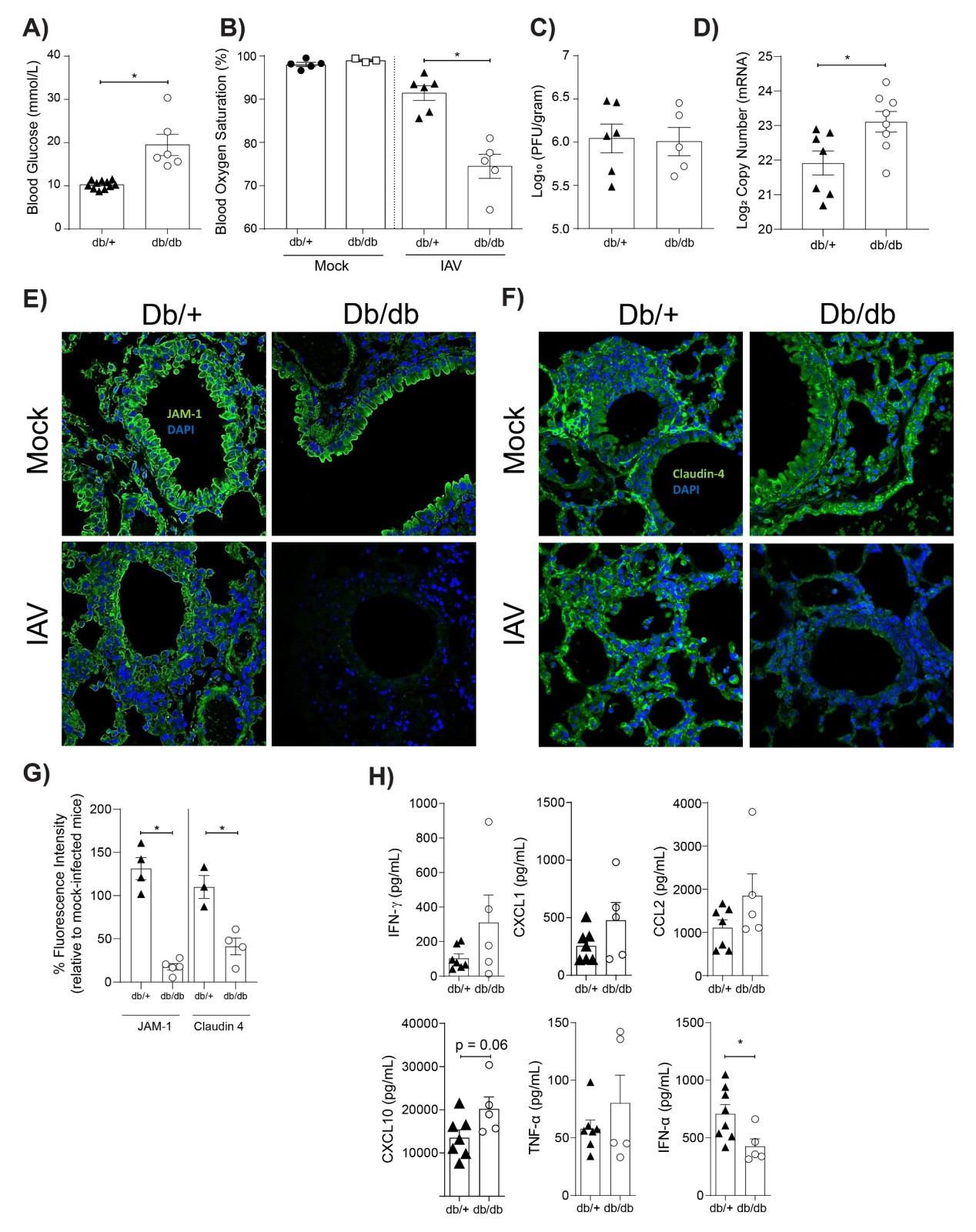

**Figure 6.** Mice with hyperglycaemia experience more severe influenza and destruction of the apical junctional complex. Heterozygous (*lepr*$^{db/+}$) and homozygous (*lepr*$^{db/db}$) mice were infected with 5.5 × 10$^3$ PFU of Auckland/09(H1N1) (n = minimum eight mice per group) or mock infected with PBS (n = minimum three mice per group). (**A**) Blood glucose concentration prior to infection was assessed. Statistical comparisons were made using a Student's unpaired t-test. (**B**) Percentage blood oxygen saturation of mock and IAV-infected mice at 3 days post-infection (d.p.i). Statistical comparisons

*Figure 6 continued on next page*

Figure 6 continued

were made using a one-way ANOVA and Tukey post hoc test. (C) Viral titres (PFU/g) in lung homogenate 3 d.p.i. Statistical comparisons were made using a Student's unpaired t-test. (D) mRNA detected by qPCR from murine lung homogenate 3 days post-IAV infection. Viral replication represented as viral copy number. Data are pooled from two independent experiments with at least three mice per group. Data are shown as mean ± SEM. Statistical significance was determined using a Student's t-test. (E) Representative fluorescent immunohistochemistry images (40x magnification) of mock and IAV-infected heterozygous (*lepr*<sup>db/+</sup>) and homozygous (*lepr*<sup>db/db</sup>) mouse lung tissue sections stained for nucleus and the tight junction protein JAM-1 (blue and green, respectively). (F) Representative fluorescent immunohistochemistry images (40x magnification) of mock and IAV-infected heterozygous (*lepr*<sup>db/+</sup>) and homozygous (*lepr*<sup>db/db</sup>) mouse lung tissue sections stained for nucleus and the tight junction protein Claudin-4 (blue and green respectively). (G) The percentage of fluorescence intensity of the relevant tight junction molecules (JAM-1 and Claudin-4) in IAV-infected heterozygous (*lepr*<sup>db/+</sup>) and homozygous (*lepr*<sup>db/db</sup>) mouse lung tissue sections relative to mock-infected counterparts (defined as 100%) at 3 d.p.i. Statistical comparisons were made using a Student's unpaired t-test *: $p < 0.05$. Data are pooled from three independent experiments (with a minimum of three biological replicates per group) and shown as mean ± SEM. JAM-1: junctional adhesion molecule-1. (H) Levels of cytokines in the lung homogenate of IAV-infected mice at 3 d.p.i. Statistical comparisons were made using a Student's unpaired t-test. Data are pooled from two independent experiments with at least three mice per group and shown as mean ± SEM. *: $p < 0.05$.

*2012*). This is of interest given that the Na,K-ATPase (located on the basolateral side of pulmonary epithelial cells) has been implicated in the pathogenesis of influenza virus (*Peteranderl et al., 2016*; *van de Sandt et al., 2017*). Altering the activity of this pump can not only facilitate pulmonary oedema, but it can also affect the formation of the junctional complex between adjacent epithelial cells (*Rajasekaran and Rajasekaran, 2003*). Therefore, the role of hyperglycaemia, Na,K-ATPase activity and influenza severity warrants further investigation.

At present, we do not know which protein component(s) secreted by endothelial cells are responsible for the barrier damage observed *in vitro* after exposure to high glucose. Our observation that barrier damage of infected and bystander cells was dependent on heat-labile factors in the cell culture supernatant, as well as the increased inflammatory response under high glucose conditions, is consistent with a causative role for cytokines in this system. Numerous different cytokines, including TNF-α, CXCL8, TNF-β, IFN-γ and IL-6, have been reported to damage the tight junctions of epithelial cells (*van de Sandt et al., 2017*; *Ohkuni et al., 2011*; *Shen et al., 2017*). Indeed, we have previously demonstrated that a combination TNF-α and IFN-γ can reduce the integrity of the epithelial barrier via inhibition of the Na,K-ATPase (*van de Sandt et al., 2017*), and it is tempting to speculate that a similar cascade of events may occur under high glucose conditions. The *in vitro* co-culture model developed herein, whilst reductionist and excluding the role that leukocytes may play in disease development, offers a relatively high-throughput model for future studies to delineate the specific role of different pro-inflammatory mediators in barrier damage. It is possible that blocking one or more of these cytokines could be used as a novel treatment option for influenza positive diabetes patients. However, traditionally such therapeutic approaches have been thwarted by inherent redundancies in cytokine signalling and inflammation, as well as the risk of increased viral shedding in treated patients (*Karawita et al., 2019*). Rather, our findings would suggest that a more beneficial therapeutic approach would be the administration of pharmacological agents to enhance the integrity of the epithelial junctional complex.

Thaiss and colleagues recently showed that hyperglycaemia, through GLUT2-dependent transcriptional reprogramming of epithelial cells, facilitated intestinal tight junction dysfunction (*Thaiss et al., 2018*). In the present study, we also observed tight junction dysfunction under high glucose conditions. However, it is important to recognise that in contrast to the work of Thaiss and colleagues, the epithelial cells in the present study were only *indirectly* exposed to differential glucose levels (i.e. differing glucose concentrations were restricted to the lower chamber of the co-culture). Moreover, unlike the study of *Thaiss et al., 2018* the endothelial cells in the present study only differed in their *history* of glucose exposure, and all *in vitro* groups were exposed to the same glucose concentration at the time of IAV infection. Therefore, the data presented here are more congruent with the so-called 'legacy effect' described in patients with diabetes. The legacy effect has been used to explain the observation that the detrimental effects of elevated glucose levels in patients persist beyond the period of hyperglycaemia (*Chen et al., 2016*; *El-Osta et al., 2008*; *Miao et al., 2014*). Why the legacy effect occurs is unclear but it has been proposed that epigenetic changes (which persist even after transient hyperglycaemia) influence the regulation of NF-κB (*Chen et al., 2016*; *El-Osta et al., 2008*; *Miao et al., 2014*). This could represent a significant

challenge to reducing the severity of influenza virus in patients with diabetes, as it would suggest that controlling one's blood glucose levels for only a limited period of time would not be sufficient to reduce the risk of disease development. Of particular concern are observations that the physiological effects of changes in blood glucose levels can persist for up to 10 years (*Holman et al., 2008*).

At present, 1 in 11 adults are living with diabetes. Within the next 20 years, this incidence is expected to increase to 1 in 10 adults (*Sainsbury et al., 2018*). Therefore, given the ever-growing prevalence of diabetes in our society it is imperative that future studies consider the role of diabetes, and more specifically hyperglycaemia, in the pathogenesis of a wide variety of different bacterial and viral pathogens.

## Materials and methods

### Cell culture

Human epithelial NCl-H441 cells (which express alveolar type II cell markers [*Ren et al., 2016*]) were obtained from the American Type Culture Collection (ATCC; Manassas, VA) and cultured in Roswell Park Memorial Institute medium (RPMI) (Gibco, Grand Island, NY) with 10% fetal bovine serum (FBS) (Sigma, St Louis, MO) and 1% penicillin-streptomycin (Lonza, Basel, Switzerland). NC1-H441 cells were used at passages 3–8. Primary human pulmonary microvascular endothelial cells (HPMEC) were obtained from Sciencell (Carlsbad, CA) and cultured in endothelial cell growth medium (Sciencell) in fibronectin coated tissue culture flasks (2 µg/cm$^2$, Sigma). HPMEC cells were used at passages 3–7, as at higher passages these cells can demonstrate senescence (*Shen et al., 1995*). Madin-Darby canine kidney (MDCK) cells were obtained from ATCC and cultured in Dulbecco modified Eagle medium (Gibco) between passages 20 and 50. All cell lines were maintained in a humidified 37°C incubator with 95% $O_2$ and 5% $CO_2$. Each cell line has no mycoplasma contamination to report.

### *In vitro* co-cultures and mono-cultures

*In vitro* co-culture and mono-culture models of the pulmonary epithelial-endothelial barrier were established as described previously (*Short et al., 2016*), whereby epithelial (NCl-H441) cells were seeded atop a transwell membrane whilst primary human endothelial cells were seeded below the transwell membrane. Once the NCI-H441 cells reached confluency, the media in both upper and lower compartments was refreshed every 12 hr. The medium of the upper compartment was refreshed using RPMI (Gibco) with 5% FBS (Sigma), 1 µM dexamethasone (Sigma) and 7 mM of glucose (Gibco), while the lower compartment was refreshed using RPMI (Gibco) with 5% FBS (Sigma) and either 7 mM glucose for the low glucose group or 12 mM glucose for the high glucose group (Gibco) until a transepithelial/endothelial electrical resistance (TEER) of greater than 800 Ω was observed and the cells were deemed ready for viral infection.

### Tube formation assay

Endothelial cells form capillary-like structures in response to angiogenic signals found in conditioned media (*DeCicco-Skinner et al., 2014*). To confirm endothelial morphology remained unchanged by RPMI media, tube formation assay was performed essentially as described previously (*DeCicco-Skinner et al., 2014*). In short, HPMECs were cultured for 4 days in either endothelial cell growth medium (Sciencell) or RPMI (Gibco) with 5% FBS (Sigma), 1% penicillin-streptomycin (Lonza) and either 7 mM glucose or 12 mM glucose (Gibco). After 4 days of culture, a Corning Costar TC-treated 96-well plate (Sigma) was coated with Cultrex Basement Membrane Extract, Type 2 (BME) (Sigma). Once BME set, cells were seeded at a density of $1.5 \times 10^3$ cells/mL in respective media. Tubes formed within 2–4 hr and were imaged using an Eclipse Ts2 inverted microscope (Nikon Instruments, Amsterdam, North Holland, the Netherlands).

### Viral strains and titrations

Influenza virus strain A/Solomon Islands/03/2006 (Solomon Islands/06; H1N1) was used to model IAV infection *in vitro*. A/Auckland/4/2009 (Auckland/09; H1N1) was used to model IAV infection *in vivo*. Virus stocks were prepared in embryonated chicken eggs as previously described (*Brauer and Chen,*

*2015*). Titres of IAV were determined by plaque assays on MDCK cells, as previously described (*Short et al., 2011*).

### *In vitro* viral infection

One hour prior to infection, the medium of the transwell plate was refreshed using RPMI (Gibco) with 2% FBS (Sigma) and 12 mM glucose (Gibco). Cells were infected or mock infected by adding A/Solomon Islands/03/2006 (Solomon Islands/06; H1N1) at multiplicity of infection (MOI) of 0.5 or phosphate buffered saline (PBS) (Gibco) respectively to the upper compartment of the transwell system. Virus was not removed after infection and cells were monitored for 24 hr post-infection.

### *In vitro* ultra-violet inactivated viral infection

A/Solomon Islands/03/2006 (Solomon Islands/06; H1N1) was irradiated with $4 \times 1$ J/cm$^2$ using a Stratagene crosslinker (UVC 500 Ultraviolet Crosslinker, 254 nm bulbs, Amersham Biosciences, Freiburg, Germany). The irradiated virus was assessed for viral titre prior to infection of the co-culture. *In vitro* viral infection was performed as previously described in Materials and methods.

### *In vitro* anakinra treatment

Where relevant, endothelial cells were treated with Kineret (anakinra) (Swedish Orphan Biovitrum, Stockholm, Sweden), an IL-1 receptor antagonist. Specifically, 1 hr prior to influenza virus infection, 10 µg/mL Anakinra was added to the lower compartment of the transwell system.

### Transepithelial/endothelial Electrical Resistance (TEER)

The TEER of the transwell cultures was monitored using an EVOM2 voltohmmeter (World Precision Instruments, Sarasota, FL, USA) with an STX2 chopstick electrode.

### Transfer assay

The *in vitro* transfer of supernatant from co-cultures was performed as described previously (*Delorme-Axford et al., 2013*). Cell culture supernatant from endothelial cells grown in co-culture was harvested 24 hr after mock or IAV infection. Harvested media was either heat inactivated for 45 min at 65°C to denature proteins or transferred without heat inactivation. Co-cultures were exposed to harvested media for 24 hr and TEER was monitored during this time.

### Lactate dehydrogenase (LDH) release

LDH release was determined using the CytoTox 96 Non-Radioactive Cytotoxicity Assay (Promega, Mannheim, Germany) as per the manufacturer's guidelines.

### FITC-dextran transport assay

Permeability to fluorescein isothiocyanate (FITC)-dextran (Sigma) was measured as described previously (*Short et al., 2016*).

### RNA extraction, cDNA synthesis and quantitative polymerase chain reaction (qPCR)

Viral and host RNA was extracted from lysed HPMEC and NCI-H441 samples, cDNA was synthesised, and real time PCR was performed as previously described (*Short et al., 2016*). Forward and reverse primer sequences for each gene of interest are shown in *Table 1*. Gene expression was normalised relative to glyceraldehyde 3-phosphate dehydrogenase (GAPDH) expression and fold change was calculated using the ∆∆Ct method (*Schmittgen and Livak, 2008*). Viral copy number was determined using IAV strain A/Puerto Rico/8/1934 H1N1 (PR8) virus matrix (M) gene cloned into pHW2000 plasmid and viral copy number was determined as described previously (*Short et al., 2013*).

### RNASeq

Sequencing quality was first assessed using FastQC to ensure that all samples had high-quality sequencing data. Reads were then aligned to Hg38 with HISAT2. Read counts were generated using HTSeq count with gencode v24 as the reference gene set. Differential gene expression was

**Table 1.** Primers used in the present study.

| Gene | Sequence |
| --- | --- |
| GAPDH | FW: CGAGATCCCTCCAAAATCAA<br>RV: TTCACACCCATGACGAACAT |
| TNF-α | FW: AGCCCATGTTGTAGCAAACC<br>RV: TGAGGTACAGGCCCTCTGAT |
| OAS1 | FW: AGAGACTTCCTGAAGCAGCG<br>RV: GAGCTCCAGGGCATACTGAG |
| IL-6 | FW: CACAGACAGCCACTCACCTC<br>RV: TTTTCTGCCAGTGCCTCTTT |
| CXCL8 | FW: GAATTGGAAAGAGGAGAGTGACAGA<br>RV: GTCTCCACACTCTTTTGGATGCT |
| Influenza A Matrix | FW: AAGACCAATCCTGTCACCTCTGA<br>RV: TCCTCGCTCACTGGGCA |

determined using EdgeR with a generalised linear model design, and an expression cut-off of two counts per million. Gene Ontology (GO) was analysed with Panther Protein Classification System to identify biological processes that were significantly enriched relative to the entire human genome.

## Mouse strains

Female and male heterozygote and homozygote $lepr^{db}$ on C57BL/6 background were kindly provided by Associate Professor Sumaira Hasnain (Mater Research Institute, Australia). Mice between the ages of 6 to 8-week old were used for all experiments. Mice were housed in individually ventilated cages, under alternating 12 hr light/dark periods with access to clean drinking water and pelletised food ad libitum. All animal experiments were approved by the University of Queensland Animal Ethics Committee (permit no. 071/17).

## Murine blood glucose measurements

Prior to infection, blood samples were collected via tail prick and blood glucose levels were measured using a SensoCard Plus blood glucose monitor (Point of Care Diagnostics, North Rocks, NSW Australia) with SensoCard test strips (Point of Care Diagnostics).

## In vivo viral infection

Mice were infected with $5.5 \times 10^3$ plaque forming units (PFU) of influenza A/H1N1/Auckland/09 intranasally in 50 μL PBS or mock-infected with PBS under isoflurane-induced anaesthesia using a Stinger Research Anesthetic Gas Machine (Darvall, Arizona). Mice were monitored daily for weight loss and clinical signs of disease. At day 3 post-infection, mice were euthanised by an intraperitoneal injection of a lethal dose of phentobarbitol (270 mg/kg) (Virbac, Millpera, Australia).

## Blood oxygen saturation

Blood oxygen saturation was measured at 3 days post-infection using a collar sensor and Mouseox Plus pulse oximeter (Starr, Oakmont, PA).

## Histology

After euthanasia, murine lungs were inflated by intratracheal administration of PBS (600 μL). The left lobe was then fixed for at least 24 hr in 10% formalin and then transferred to 70% ethanol for processing by the Core Histology Facility, Translational Research Institute. Samples were imbedded in paraffin wax and sliced to a thickness of 5 μm using a Hyrax M25 Rotary Microtome (Leica Biosystems, Nussloch, Germany).

## Immunofluorescence

Cells grown on a transwell membrane were fixed, stained and mounted on a glass slide with mounting medium containing 4′,6-diamidino-2-phenylindole (DAPI) (Vector Laboratories, Burlingame, CA) as previously described (*Short et al., 2016*). Primary antibodies used were α-E-cadherin (Bioss

Company, Beijing, China), α-junctional adhesion molecule 1 (Santa Cruz Biotechnology, Santa Cruz, CA), α-claudin-4 (Life Technologies, Gaithersburg, MD) and a monoclonal antibody (clone HB65 IgG2a) directed against the nucleoprotein of influenza A virus (American Type Culture Collection, Manassas, VA). Secondary antibodies were Alexa 488 α-mouse IgG2a, Alexa 647 α-mouse IgG2a and Alexa 488 α-rabbit IgG (Life Technologies).

Unstained murine lung paraffin sections were de-waxed and rehydrated by a series of xylene and ethanol washes. Slides were then incubated in proteinase K (2 mg/mL) for 10 min at 37°C and subject to a series of PBS and PBS/0.01% Tween-20. Slides were incubated with the relevant primary antibody for 1 hr at room temperature followed by PBS washes. Slides were next treated with 0.01% Trypan blue (Sigma) for 3 min at room temperature. Slides were then incubated for 1 hr at room temperature using the relevant secondary antibody. Sections were subsequently washed with PBS and mounted with mounting medium containing DAPI (Vector Laboratories). All antibodies were diluted in PBS/1% bovine serum albumin (Sigma).

Immunolabelling of transwell membranes and murine lung histological sections were visualised using a Laser Scanning Microscope 710 (LSM 710) (Zeiss, Jena, Germany). Staining intensity of tight junction proteins was quantified using ImageJ software (National Institutes of Health, Bethesda, MD). Specifically, an imageJ macro was created to measure the mean grey value within a 6 cm$^2$ area. Each measurement was taken of an image section with 8–12 nuclei sitting within this 6 cm$^2$ area of transwell confocal images. Three measurements were taken per image, and three images were quantified per transwell membrane. Quantification of murine lung tight junction signal intensity was performed using the same methodology.

## Cytokine levels

Cytokines in the clarified murine lung homogenate, HPMEC and NCl-H441 co-culture supernatant were measured using the LEGENDplex anti-virus response panel according to the manufacturer's instructions (Biolegend, San Diago, CA). Outliers were identified and excluded using Grubbs' test.

## Statistical analysis

Statistical analyses were performed using Graph Pad Prism software (version 7.02) for Windows (GraphPad Software, La Jolla, CA).

## Acknowledgements

This work was supported by a NHMRC Project grant (APP1159959). LAG is supported by an Early Career Fellowship from the National Health and Medical Research Council and Heart Foundation (Australia). KRU is supported by a NHMRC Career Development Fellowship (APP1130815). KRS is supported by an Australian Research Council DECRA (DE180100512). Kineret (anakinra) was kindly provided by Professor Kate Schroder.

## Additional information

### Funding

| Funder | Grant reference number | Author |
| --- | --- | --- |
| National Health and Medical Research Council | APP1159959 | Linda A Gallo Kirsty R Short |
| National Health and Medical Research Council | Early-Career Research Fellowship | Linda A Gallo |
| National Health and Medical Research Council | APP1130815 | Kyle R Upton |
| Australian Research Council | DE180100512 | Kirsty R Short |

The funders had no role in study design, data collection and interpretation, or the decision to submit the work for publication.

## Author contributions
Katina D Hulme, Data curation, Formal analysis, Validation, Investigation, Visualization, Methodology, Writing - original draft, Project administration, Writing - review and editing; Limin Yan, Kyle R Upton, Formal analysis; Rebecca J Marshall, Conor J Bloxham, Keng Yih Chew, Data curation; Sumaira Z Hasnain, Zhixuan Loh, Resources; Helle Bielefeldt-Ohmann, Data curation, Formal analysis; Katharina Ronacher, Linda A Gallo, Supervision; Kirsty R Short, Conceptualization, Resources, Supervision, Funding acquisition, Investigation, Writing - original draft, Writing - review and editing

## Author ORCIDs
Katina D Hulme  https://orcid.org/0000-0003-1322-0136
Kirsty R Short  https://orcid.org/0000-0003-4963-6184

## Ethics
Animal experimentation: All animal experiments were approved by the University of Queensland Animal Ethics Committee. (permit no. 071/17).

## Decision letter and Author response
Decision letter https://doi.org/10.7554/eLife.56907.sa1
Author response https://doi.org/10.7554/eLife.56907.sa2

# Additional files

## Supplementary files
• Transparent reporting form

## Data availability
RNA sequencing data generated in this study are available on the Gene Expression Omnibus repository with accession number GSE145232 (https://www.ncbi.nlm.nih.gov/geo/query/acc.cgi?acc=GSE145232).

The following dataset was generated:

| Author(s) | Year | Dataset title | Dataset URL | Database and Identifier |
|---|---|---|---|---|
| Short KR, Upton KR | 2020 | High glucose levels increase influenza-associated damage to the pulmonary epithelial-endothelial barrier | https://www.ncbi.nlm.nih.gov/geo/query/acc.cgi?acc=GSE145232 | NCBI Gene Expression Omnibus, GSE145232 |

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
