## [Decision Letter]

**Acceptance summary:**

The major points of the study are that the presence of high concentrations of glucose in blood (as in uncontrolled diabetes) increases the influenza virus-associated damage to lung tissue at the level of the epithelial-endothelial barrier. Degradation of the tight junctions between epithelial cells in the lung is part of this pathology. This study provides a potential explanation for the severity of influenza in patients with diabetes, and suggests that blood glucose control might contribute to a more favourable course of influenza in these patients.

**Decision letter after peer review:**

Thank you for submitting your article "High glucose levels increase influenza-associated damage to the pulmonary epithelial-endothelial barrier" for consideration by *eLife*. Your article has been reviewed by three peer reviewers, one of whom is a member of our Board of Reviewing Editors, and the evaluation has been overseen by Jos van der Meer as the Senior Editor. The reviewers have opted to remain anonymous.

The reviewers have discussed the reviews with one another and the Reviewing Editor has drafted this decision to help you prepare a revised submission.

Summary:

Clinically, it is well established that diabetes increases the risk of developing severe influenza infections. In this manuscript Dr. Short and colleagues characterize the effect of high glucose levels on the integrity of the pulmonary epithelial- endothelial barrier in an *in vitro* system as well as in an *in vivo* mouse model. Exposure of the endothelial cell layer to higher levels of glucose results in more pronounced decreases of transepithelial/endothelial electrical resistance (TEER) upon influenza A (IAV) infection. This seemingly occurs without differences in replication efficiency. Higher glucose levels lead to a more pronounced innate immune response in the endothelial cells, and the secreted cytokines might underlie the damage to the epithelial layer, as assessed by reduced protein levels of markers for the apical junctional complex. The study further establishes a (testable) mechanisms that supports the observation of higher mortality of IAV infection in diabetes. Overall, the work is well performed, timely and the conclusions are largely supported by the dataset. There are few areas that need to be clarified.

1) The proposed mechanisms that endothelial cell injury precedes the epithelial cell damage is well supported by the *in vitro* data, however, not been causally linked *in vivo*. Additional evidence on endothelial cell injury, for example by immunofluorescence localization and earlier time points would significantly substantiate this conclusion. Please provide additional data or adapt the claims accordingly.

2) It will be important to shed more light on whether the initial effects are mediated through active viral replication or innate immune responses. The authors mention the discrepancy with previous data where the effect of glucose (Kohio and Adamson, 2013) was attributed to higher levels of replication, which on itself could attribute to enhanced damage to the epithelial cell tight junctions (Short et al., 2016). The authors show at a late time point that viral replication (PFU and viral RNA levels) is not different at high and low glucose *in vitro*. Does this mean that viral RNA replication is not important for the difference in TER and the effects are perhaps due to innate immune responses rather than direct damage of viral gene products? I noticed that the difference between the low and high glucose in TER is already significant at 6hpi (Figure 1B) and the magnitude of the difference does not become much more at later time-points. This suggests that it is quite an immediate effect and perhaps does not require viral replication to occur. To shed more light on the apparent discrepancy of viral replication it would be important to test if exposure to UV-inactivated IAV leads to a similar difference in TER between high and low glucose.

3) The data presented suggests that the difference between high and low glucose conditions on the epithelial cell layer for a large part can be explained by cytokines secreted by endothelial cells going against a model where viral replication (more expression of viral gene products) directly does the damage. Yet, *in vivo* there is higher viral mRNA accumulation, which still leaves the viral replication hypothesis open. Of course it could also be that loss of integrity requires both replication and cytokines. A co-staining in Figure 4B of the tight junction marker with a viral gene product could clarify whether loss of integrity only occurs in infected cells or whether also bystanders are affected.

4) The heat labile media activity may be due to IL-1 – given its ability to disrupt TEER? Does Anakinra or anti-IL1 disrupt this activity? This needs to be verified.

Revisions expected in follow-up work:

1) The *in vitro* model depends on a cancer cell line (epithelial). Would improve the manuscript significantly in terms of relevance by inclusion of human primary cells, which are available and well-studied. Key findings should be confirmed.

---

## [Author Response]

Revisions for this paper:1) The proposed mechanisms that endothelial cell injury precedes the epithelial cell damage is well supported by the *in vitro* data, however, not been causally linked *in vivo*. Additional evidence on endothelial cell injury, for example by immunofluorescence localization and earlier time points would significantly substantiate this conclusion. Please provide additional data or adapt the claims accordingly.

Thank you for your suggestion of earlier time points for the *in vivo* experiment to further substantiate our conclusions. Due to COVID-19 we no longer have access to the strain of mice required for these experiments and thus and cannot perform earlier time points than the D3 provided in the manuscript. We would also like to remind the reviewer that rather than endothelial cell injury and death, we hypothesise that it is a pro-inflammatory state of the endothelial cells that is driving the phenotype. Consistent with this notion, using activated caspase 3 staining on the *in vivo* sections, we can confirm that there is no difference in endothelial cell death (Author response image 1). To clarify this point further we have now altered the revised manuscript to read: “Whilst we cannot offer definitive evidence that these cytokines were endothelial cell derived (as was the case with the *in vitro* data), these data are consistent with our *in vitro* data that loss of intact tight junctions in the context of IAV and hyperglycaemia is associated with a pro-inflammatory response”.

**Author response image 1. sa2fig1:** Histopathology scoring of lung sections for activated caspase 3. Data are pooled from two independent experiments, with mean ± SEM.

2) It will be important to shed more light on whether the initial effects are mediated through active viral replication or innate immune responses. The authors mention the discrepancy with previous data where the effect of glucose (Kohio and Adamson, 2013) was attributed to higher levels of replication, which on itself could attribute to enhanced damage to the epithelial cell tight junctions (Short et al., 2016). The authors show at a late time point that viral replication (PFU and viral RNA levels) is not different at high and low glucose *in vitro*. Does this mean that viral RNA replication is not important for the difference in TER and the effects are perhaps due to innate immune responses rather than direct damage of viral gene products? I noticed that the difference between the low and high glucose in TER is already significant at 6hpi (Figure 1B) and the magnitude of the difference does not become much more at later time-points. This suggests that it is quite an immediate effect and perhaps does not require viral replication to occur. To shed more light on the apparent discrepancy of viral replication it would be important to test if exposure to UV-inactivated IAV leads to a similar difference in TER between high and low glucose.

This is an interesting query and we thank the reviewer for suggesting it. We have now performed additional *in vitro* experiments using UV-inactivated virus (Figure 3—figure supplement 1). UV inactivated virus did not induce barrier damage, even under hyperglycaemic conditions, indicating that active virus is required to induce a reduction in TEER. In our manuscript we suggest that it is a soluble protein (such as a cytokine) produced in response to IAV infection that is driving barrier damage. Therefore, these data may suggest that the production of this soluble factor is dependent on active viral replication. Alternatively, it is possible that UV-inactivated virus does not active PRRs in the same manner that active replicating virus does, thus resulting in a differential immune response.

We have included these data in Figure 3—figure supplement 1 of the revised manuscript and a discussion of these points in the Results: “Furthermore, infection using ultraviolet inactivated virus did not induce barrier damage, even under hyperglycaemic conditions, indicating that initially, active virus is required to induce a reduction in TEER (Figure 3—figure supplement 1)”.

In the Discussion: “Interestingly, in the present study, we observed that hyperglycaemia increased IAV-induced barrier damage *in vitro* and increased influenza severity *in vivo* in the absence of increased viral replication (although initial viral replication was required to trigger epithelial barrier damage)”.

Additionally: “Our observation that barrier damage of infected and bystander cells was dependent on heat-labile factors in the cell culture supernatant, as well as the increased inflammatory response under high glucose conditions, is consistent with a causative role for cytokines in this system”.

The Materials and methods section has been adjusted to include: “*In vitro* ultra-violet inactivated viral infection. A/Solomon Islands/03/2006 (Solomon Islands/06; H1N1) was irradiated with 4x1J/cm2 using a Stratagene crosslinker (UVC 500 Ultraviolet Crosslinker, 254nm bulbs, Amersham Biosciences, Freiburg, Germany). The irradiated virus was assessed for viral titre prior to infection of the co-culture. *In vitro* viral infection was performed as previously described in the Materials and methods”.

3) The data presented suggests that the difference between high and low glucose conditions on the epithelial cell layer for a large part can be explained by cytokines secreted by endothelial cells going against a model where viral replication (more expression of viral gene products) directly does the damage. Yet, *in vivo* there is higher viral mRNA accumulation, which still leaves the viral replication hypothesis open. Of course it could also be that loss of integrity requires both replication and cytokines. A co-staining in Figure 4B of the tight junction marker with a viral gene product could clarify whether loss of integrity only occurs in infected cells or whether also bystanders are affected.

We thank the reviewer for this comment. We have now performed a co-staining for influenza virus antigen and tight junctions as the reviewer has suggested (**Figure 4D**). These results show there is a reduction in tight junction integrity in uninfected cells, suggesting that a loss of integrity is not restricted to infected cells. The role of viral replication vs. inflammation in the observed phenotype is an interesting question. Based on the new data generated using the UV virus (reviewer question 2), we would hypothesise that initial viral replication in epithelial cells triggers the release of a soluble factor from endothelial cells that is then responsible for epithelial barrier damage (independent of whether the cell was virally infected or not). Accordingly, the phenotype is dependent upon active virus (as shown with the UV experiments) but barrier damage occurs in both infected and bystander cells (demonstrated in Author response image 2).

We have now included a discussion of this proposed model in the Results: “Consistent with our hypothesis that a soluble factor is driving the loss of barrier integrity, co-staining of Claudin-4 and the influenza A nucleoprotein demonstrate that barrier damage occurs in both infected and bystander cells (Figure 4D)”. As well as: “Our observation that barrier damage of infected and bystander cells was dependent on heat-labile factors in the cell culture supernatant, as well as the increased inflammatory response under high glucose conditions, is consistent with a causative role for cytokines in this system”.

Similarly, the Materials and methods section has been updated to reflect this: “Primary antibodies used were α-E-cadherin (Bioss Company, Beijing, China), α-junctional adhesion molecule 1 (Santa Cruz Biotechnology, Santa Cruz, CA, USA), α-claudin-4 (Life Technologies, Gaithersburg, MD, USA) and a monoclonal antibody (clone HB65 IgG2a) directed against the nucleoprotein of influenza A virus (American Type Culture Collection, Manassas, VA, USA). Secondary antibodies were Alexa 488 α-mouse IgG2a, Alexa 647 α-mouse IgG2a and Alexa 488 α-rabbit IgG (Life Technologies).”

**Author response image 2. sa2fig2:** Schematic representation of hypothesised mechanism of tight junction barrier damage under hyperglycaemic conditions *in vitro*. Active replicating virus triggers signalling either directly or indirectly acting on the endothelial cells (1). Endothelial cells exposed to a history of hyperglycaemia produce a heat-labile soluble protein into the supernatant (3). This soluble factor acts upon the tight junction proteins of the epithelial cells, causing degradation and barrier permeability in both infected and non-infected epithelial cells (4). Image created with Biorender.

4) The heat labile media activity may be due to IL-1 – given its ability to disrupt TEER? Does Anakinra or anti-IL1 disrupt this activity? This needs to be verified.

In response to the reviewer’s query, we have now performed experiments to block IL-1 activity using Anakinra and found that the TEER disruption seen in the 12mM condition is independent of IL-1 activity. We have now included this as a supplementary figure in the updated manuscript (Figure 3—figure supplement 2) and have updated the text in the Results: “Il-1β is known to increase interstitial epithelial barrier permeability (Al-Sadi and Ma, 2007). […] These results are to be expected due to the redundancy in innate host defence pathways for pathogen recognition (Nish and Medzhitov, 2011)”.

The Materials and methods section has also been updated: “*In vitro*Anakinra treatment. Where relevant, endothelial cells were treated with Kineret (anakinra) (Swedish Orphan Biovitrum, Stockholm, Sweden), an IL-1 receptor antagonist. Specifically, one hour prior to influenza virus infection, 10µg/mL Anakinra was added to the lower compartment of the transwell system”. Additionally, the acknowledgments have been updated: “Kineret (anakinra) was kindly provided by Professor Kate Schroder”.

Revisions expected in follow-up work:1) The *in vitro* model depends on a cancer cell line (epithelial). Would improve the manuscript significantly in terms of relevance by inclusion of human primary cells, which are available and well-studied. Key findings should be confirmed.

We thank the reviewer for their comment and agree with the suggestion. We are currently culturing primary human epithelial cells that we endeavour to use for such experiments.